# Prevalence of Violence against Providers in Heart and Lung Transplant Programs

**DOI:** 10.3390/ijerph20064805

**Published:** 2023-03-09

**Authors:** Todd A. Barrett, Gennaro Di Tosto, Karen Shiu-Yee, Halia L. Melnyk, Laura J. Rush, Lindsey N. Sova, Brent C. Lampert, Asvin M. Ganapathi, Bryan A. Whitson, Brittany L. Waterman, Ann Scheck McAlearney

**Affiliations:** 1Division of Palliative Medicine, The Ohio State University Wexner Medical Center, Columbus, OH 43210, USA; 2The Center for the Advancement of Team Science, Analytics, and Systems Thinking in Health Services and Implementation Science Research (CATALYST), College of Medicine, The Ohio State University, Columbus, OH 43210, USA; 3Heart and Vascular Center, Richard M. Ross Heart Hospital, The Ohio State University Wexner Medical Center, Columbus, OH 43210, USA; 4Division of Cardiovascular Medicine, The Ohio State University Wexner Medical Center, Columbus, OH 43210, USA; 5Division of Cardiac Surgery, The Ohio State University Wexner Medical Center, Columbus, OH 43210, USA; 6Department of Family and Community Medicine, College of Medicine, The Ohio State University, Columbus, OH 43210, USA

**Keywords:** transplant, transplantation, workplace violence, healthcare workers

## Abstract

Workplace violence in healthcare institutions is becoming more frequent. The objective of this study was to better understand the nature of threat and physical acts of violence from heart and lung transplant patients and families toward healthcare providers and suggest programmatic mitigation strategies. We administered a brief survey to attendees at the 2022 International Society of Heart and Lung Transplantation Conference in Boston, Massachusetts. A total of 108 participants responded. Threats of physical violence were reported by forty-five participants (42%), were more frequently reported by nurses and advanced practice providers than physicians (67% and 75% vs. 34%; *p* < 0.001) and were more prevalent in the United States than abroad (49% vs. 21%; *p* = 0.026). Acts of physical violence were reported by one out of every eight providers. Violence against providers in transplant programs warrants closer review by health systems in order to ensure the safety of team members.

## 1. Introduction

Workplace violence directed toward healthcare providers encompasses all “incidents where staff are abused, threatened or assaulted in circumstances related to their work […], involving an explicit or implicit challenge to their safety, well-being or health” [1,2]. It is an important concern that has been exacerbated by the COVID-19 pandemic [2,3,4,5,6,7]. Recent systematic reviews and meta-analyses show that between 57–67% of the workforce experiences verbal abuse and 21–33% experiences threats of physical violence [2,5]. In the U.S., healthcare workers are five times more likely to experience workplace violence injuries than workers from all industries [8]. Non-fatal workplace injuries and illnesses due to violence perpetrated upon healthcare workers accounted for 76% of all workplace injuries in 2020 [9]. Workplace violence, at its worst, results in fatalities such as in the case of an orthopedic surgeon in Tulsa, Oklahoma, who was killed in a mass shooting by a disgruntled former patient [10]. Non-fatal violence incidents have personal and occupational consequences such as emotional distress, poor health behaviors, psychiatric disorders, and diminished job satisfaction/performance [3]. Despite the high prevalence of healthcare workplace violence, there are aspects of the problem that remain unaddressed and understudied [3]. Issues requiring more attention include the underreporting of incidents and lack of clear guidance for practitioners on what and how to report regarding these incidents [3,11,12]. There is also a need to collect different data elements (e.g., provider gender) [5] and harmonize validated data collection tools/instruments to comprehensively study workplace violence [3]. Additionally, a better understanding of the type of settings in which there is a higher risk for workplace violence is needed in order to help fashion appropriate interventions [3,13].

Heart and lung transplantation programs are a unique hospital setting with regard to the potential for workplace violence [14]. There are two primary reasons why threats of violence or actual violence from transplant patients in these programs and their families may pose challenges to providers that are distinct from those of other settings. First, heart and lung transplant patients often have unpredictable disease trajectories that can result in complex disease burden and/or sudden death [15,16]. Second, given the limited number of heart and lung transplantation programs [17], referral to another transplant center when patients and/or their families threaten violence or become physically violent can be difficult or impossible. Despite the potential for increased workplace violence in heart and lung transplantation program settings, there is scarce information in the literature about how commonly providers in this field encounter violence. To better characterize the prevalence of threats and acts of physical violence against heart and lung transplant providers from patients and their families, we surveyed providers working in heart and lung transplantation programs across the world and asked about their experiences.

## 2. Materials and Methods

### 2.1. Study Setting and Population

This cross-sectional, online survey was approved by the Institutional Review Board of The Ohio State University. Participants were recruited by four members of the research team (T.A.B., B.A.W., B.C.L. and A.M.G.) while in attendance at the 42nd Annual Meeting of the International Society of Health and Lung Transplantation held in Boston, Massachusetts from 27–30 April 2022. Participants were drawn from a convenience sample of conference attendees: four providers from the research team approached conference attendees gathered in small groups between sessions and promoted the study. Potential respondents were provided with a brief explanation of the goal of the study and given access to the online survey via a quick response (QR) code. Due to the informal recruiting strategy, the research team did not attempt to keep track of the number of attendees they interacted with during the conference. All respondents provided informed consent prior to participation (see Appendix A for consent form).

### 2.2. Survey Measures

The survey contained ten items. Three of the survey questions asked respondents about basic program characteristics and demographics: program type (heart transplant, lung transplant, both); role (physician, advanced practice provider, nurse, social worker, other); and geographic location (Midwest, Northeast, South, West, International). Six of the survey questions asked for respondents’ perceptions about physical violence from patients and families. An open-ended question inviting respondents to provide additional relevant comments concluded the survey. Respondents were not asked to provide any personal information.

### 2.3. Statistical Analysis

There were 120 conference attendees who consented to study participation. Twelve did not respond to the questions asking about the prevalence of physical violence and were therefore excluded from the analysis. Of the remaining 108 respondents, 83 (77%) answered all survey questions. Incomplete responses were left in the analytical sample. Missing answers were not imputed and were handled via pairwise deletion after we determined that the data was missing at random.

Survey items were summarized using frequencies and percentages. Answers to the questions about threats and acts of physical violence were analyzed in relation to the characteristics of the respondent’s clinic, their role in the clinic, and their geographic location. Geographic information about U.S. respondents was grouped into regions according to the U.S. Census Bureau [18]. Pearson’s Chi-squared tests were used to study differences between groups. In the cases of the respondents’ roles and locations where tests had more than one degree of freedom, standardized residuals with an absolute value greater than two were used as indication that the null hypothesis of independence was violated for that group [19]. All analyses were performed in R version 4.1.2, with two-sided *p* < 0.05 considered significant.

## 3. Results

### 3.1. Characteristics of Study Participants

One hundred and eight participants were included in the analytical sample (Table 1). Fifty-one percent of participants worked in heart transplantation programs, 36% in lung transplantation programs, and 13% in programs that perform both lung and heart transplants. A majority (65%) of participants were physicians; the sample also included 17% advanced practice providers and 11% nurses. Participants included providers from each region of the U.S., with 20% of the sample reporting that they practiced abroad.

### 3.2. Prevalence of Workplace Violence

Forty-two percent of participants reported they or a team member had experienced a threat of physical violence from a patient or family member, and 12% reported they or a team member had experienced actual physical violence (Table 2). In response to the open-ended question, one respondent further disclosed that they were “punched in the face,” and another commented that a “cardiac surgeon at [their] hospital was killed by gunshot in the clinic by a disgruntled family member of a deceased patient.” When asked about other transplantation teams, 55% of respondents said they knew of a provider who had been threatened with physical violence, and 25% of participants reported they knew of a transplantation provider who had experienced physical violence carried out by a patient or family member. Of note, despite this reported prevalence of violence, nearly all participants (99%) said they felt safe at work either “most of the time” or “all of the time.” When asked how they felt physical violence or the threat of physical violence in their practice has changed over time, most (67%) participants felt it had stayed the same while 30% reported feeling it had increased.

### 3.3. Geographic Variation in Workplace Violence

Comparisons of providers’ reported experiences with workplace violence by geographic location are reported in Table 3. We found significantly more providers in the U.S. had received threats of physical violence from patients and families than had providers from abroad (*p* = 0.026). The proportion of providers who suffered from acts of physical violence, however, was similar both among U.S. and international providers (*p* = 0.2). Within the U.S., there were no statistically significant differences by region in providers’ experiences with threats of physical violence from patients and families (Table 4). Yet here was a significant regional difference in the prevalence of acts of physical violence. In the Western U.S., 40% of respondents reported that they or a team member had been a victim of physical acts of violence (*p* = 0.01).

### 3.4. Variation by Provider Role

The prevalence of violence from patients or their families was significantly different based on provider role (Table 5). A higher percentage of nurses (67%) and advanced practice providers (75%) reported threats of physical violence compared to their physician counterparts (*p* < 0.001). Additionally, a higher percentage of nurses (24%) and advanced practice providers (33%) responded that they or a team member of theirs had suffered from acts of physical violence from patients or families relative to physician participants (*p* = 0.014). Interestingly, many respondents across all provider roles reported feeling safe “most of the time” or “always”.

## 4. Discussion

Our study found that one out of every eight participating heart and lung transplant providers stated that they or a member of their team had been physically assaulted in the past. Additionally, our study found that the nursing and advance practice professions reported more experiences with physical violence and threats of physical violence than did physician providers. It is not surprising that heart and lung transplantation programs have not escaped this phenomenon, given that healthcare is the profession with the highest rate of workplace violence [8]. Key decision makers on transplant teams may be unaware of the prevalence of violence in their programs due to inconsistent reporting and underreporting of such threats and acts. 

### 4.1. Unique Challenges for Transplant Programs

Transplantation programs face unique challenges when confronted with workplace violence. Heart and lung transplantation programs are located at quaternary care centers across the world. In the U.S., there are approximately 200 programs [17] to serve a population of more than 331,000,000 people [20]. When faced with assault or the threat of physical violence, the process of discharging a patient from a transplant program is complex. Referring patients to another program may not be possible for a few reasons. First, the patient may not have the ability to travel the distance needed to be seen by another program. Moreover, some areas of the country may not have another transplantation center that exists or that accepts the patient’s insurance. Second, the alternative transplant center may not be willing to accept a patient into their longitudinal program when they have been discharged due to violence. If a patient has recently received a transplant, programs will be held accountable for the patient’s outcomes despite the patient’s history of threats or physical act of violence toward providers. This creates a programmatic barrier to early discharge and referral to an alternative care location. 

### 4.2. Risk Factors for Workplace Violence

There are several known risk factors for the commission of workplace violence from patients and families. A primary risk factor is mental health disorders [21,22,23]. While fields such as psychiatry and emergency medicine may have proportionately more patients with mental health disorders, there is evidence of a high prevalence of depression and anxiety *among heart and lung transplant patients [24]. If left untreated, these conditions are contraindications for transplantation [25]. Notably, of all the outpatients at our medical center in 2021 who were screened as potential candidates for heart transplant using the Hospital Anxiety and Depression Scale [26], 22% were borderline abnormal or abnormal for depression, and 40% were borderline abnormal or abnormal for anxiety (unpublished data, March 2022). Our data underscore that mental health issues are important to consider and treat in order to reduce the risk for violence in transplantation programs. 

Another known risk factor for workplace violence is the inability for a patient or family member to cope with crisis situations [27]. Patients undergoing heart or lung transplantation are subject to extreme stress. In fact, 25% of patients on the waiting list for lung transplantation screen positive for post-traumatic stress disorder (PTSD) [28], and 14% of heart transplant recipients have PTSD symptoms following their transplant procedure [29]. The acute, heightened stress around heart and lung transplantation decision-making can contribute to elevated risks of physical violence and the threat of physical violence against transplant providers [21] if these patient- and family-level factors are not addressed.

Interestingly, our respondents reported feeling safe at work all or most of the time, despite reporting a high prevalence of violence or threats of violence. More research is needed to understand the reasons for this finding. Specifically, it is possible that healthcare workers have come to accept violence as another facet of their many workplace challenges [30], or that their feelings of safety are the result of their institutions having adequate protective measures in place.

However, it is known that workplace violence is often underreported in the healthcare setting [11,12]. It has been shown that in addition to underreporting, healthcare workers often have inconsistent guidance on how to accurately report episodes of violence [3]. Our findings add context to the series of high-profile murders of healthcare providers that have taken place in recent years [10,31,32,33]. Furthermore, violent crime in the U.S. has risen sharply, from a rate of 361 per 100,000 people in 2014 to 398.5 per 100,000 in 2020 [34]. Additionally, gun violence in the U.S. has climbed steadily, resulting in 45,222 deaths in 2020; this is a 14% increase from 2019 and a staggering 43% increase from 2010 [35]. Taken together, these findings make it clear that the risk of violence against healthcare providers is not theoretical, and steps need to be taken to ensure the safety of all healthcare team members.

### 4.3. Recommendations for Transplant Programs

Considering these findings and the growing body of evidence about violence and workplace safety, it is important for transplant programs to acknowledge the real threat of workplace violence, assess their local context, and develop strategies to ensure the safety of their teams. Based on our findings, the existing literature on workplace violence in health care settings in general, and our provider experience in a major heart and lung transplantation program, we therefore propose the following seven steps for all transplant programs.

#### 4.3.1. Set Expectations Pre-Transplant

Much like existing opioid contracts or inpatient care behavioral contracts for disruptive patients, transplantation programs should consider developing behavioral contracts with all patients that include a zero-tolerance policy for workplace violence [36]. Individual programs can determine the criteria for dismissing patients from their transplantation program and clearly present these expectations to transplantation candidates in a behavioral contract presented pre-transplant.

#### 4.3.2. Behavioral Modification and Safety Plans

Transplantation teams should develop a standardized risk-reduction safety strategy for patients and family members who have threatened physical violence [37] but have not yet been discharged or terminated from the program. Involving hospital security and administration is critical so the plan can be authorized and enacted quickly if necessary. Such a plan may include weapon searches, security escorts for patient visits, clearly flagging the medical record to notify team members, and removing team members involved in the threat [27]. Additionally, given the overall low quality of the evidence regarding interventions to minimize aggression directed toward health care workers by patients [37], more research is needed to develop patient-facing tools to minimize distress and maximize the outcomes of those awaiting heart and lung transplants.

#### 4.3.3. Escalating Security and Active Shooter Training

Transplant teams need to work with their institutions to understand how to escalate security measures during an unsafe situation in both inpatient and outpatient settings. First, a workplace violence safety protocol needs to be developed, as recommended by the Occupational Safety and Health Administration [21,36]. This protocol needs to be disseminated to all members of the team. All members of the team should feel empowered to activate the plan when they feel unsafe.

#### 4.3.4. Develop a Safe Strategy for Communicating Distressing Information

Several surveys have found that workplace violence is associated with workplace stress [38]. Providers in transplant programs routinely face stressful interactions with patients and their families surrounding the communication of distressing information such as declining the patient for transplant, removing a patient from the transplantation wait list, or communicating information about a significant negative outcome. Transplantation teams, therefore, should develop a protocol for disseminating distressing information to patients and families. Having multiple team members present for this communication with patients and families is important. It can also serve to ensure that the transplant team feels empowered to escalate security when necessary.

#### 4.3.5. Develop Efficient and Clear Reporting Strategies

Transplantation teams should also attempt to reduce the burden and stigma associated with reporting workplace violence. All team members should be encouraged to report workplace violence through their quality and safety reporting systems. Furthermore, teams need to ensure that follow-up actions are taken after incident reports are filed. Actions such as notifying staff of any policy changes and/or recommendations to prevent future incidents would reinforce the importance of using internal mechanisms for reporting workplace violence [11].

#### 4.3.6. Develop Reciprocal Program Relationships

Although we found that a significantly higher proportion of U.S. providers experienced threats of physical violence than their provider counterparts abroad, there were no statistically significant difference in the US by region in terms of providers’ experiences with threats of physical violence from patients and families. Given this preliminary finding, more research is needed to determine if teams in close geographic proximity, through a process of communication and seeking to develop reciprocal relationships, could facilitate efficient referral of violent patients to alternate programs.

#### 4.3.7. Revise Quality Reporting Standards

National and international quality and safety reporting standards for transplant programs, in our opinion, should consider not including patients who are discharged due to threats or acts of violence when calculating the overall program quality score. Programs should not be penalized for protecting their workers from violence. Furthermore, the discharge of violent patients from programs should not be discouraged because of its potential to negatively impact quality scores if retaining such a patient places providers at risk of harm.

### 4.4. Limitations

Our study has several limitations. First, this was a relatively small convenience sample of providers which may limit the generalizability of our findings. In addition, it is possible that the QR scanning method of recruitment may have led to age bias in our study if younger assembly participants who were actively using technology in communal spaces were more likely to participate. Third, as we did not ask for demographic information from participants, we were unable to perform subgroup analyses such a by gender or age. While most studies of workplace violence do not report provider gender separately [5], we found one study in which there was no gender difference in acts of violence against physicians in their later years [39]. Moreover, questions did not ask about the length of time a provider had been working in a particular program, nor about the length of time to consider when reflecting upon violence in transplant programs. Lastly, given the anonymity of the survey, we were unable to track whether respondents from the same institution/program reported the same incidents of threat or violence.

## 5. Conclusions

In summary, we found that nearly half of the participants in our survey reported that they or a coworker had experienced the threat of physical violence and 12% experienced actual acts of violence. Additionally, we found that a significantly higher proportion of U.S. providers experienced threats of physical violence than their international counterparts. The proportion of providers that reported experiencing acts of violence, however, did not differ by geographic region. Lastly, we found that the prevalence of violence based on provider role differed, with a higher percentage of nurses and advanced practice providers reporting threats of physical violence and acts of physical violence compared to physicians. This is the first attempt, to our knowledge, to help to quantify the problem in the setting of heart and lung transplant programs.

Our study fills an important gap in the evidence by adding heart and lung organ transplantation programs to the list of settings in which workplace violence is very likely to occur. We discuss the reasons for an increased likelihood of violence against transplant program providers and highlight ways to mitigate the risk for violence in this setting. Decreasing the prevalence of threats and acts of violence is an important step toward increasing workforce safety. It would also help to reduce provider burnout and maintain a stable and effective workforce. Comparative study of existing organizational reporting systems could help to standardize the tracking and collection of workplace violence incidents and their outcomes. Given the dearth of information on prevention of violence in healthcare settings in general and specific to heart and lung transplant programs, a variety of approaches to ameliorating the problem should be considered. Organizational strategies would ideally include assessment and identification of staff and patient risk factors for workplace violence [40] and institution of staff education, early intervention, and support programs [13,21,36]. The involvement of professional associations in promoting legislation aimed at protecting health care workers from violence [7] and curbing the societal epidemic of gun violence is also vital to addressing this urgent and global problem.

## Figures and Tables

**Table 1 ijerph-20-04805-t001:** Respondent Characteristics.

Survey Item	N = 108 ^1^
In what type of practice do you work?	
Heart transplant	55 (51%)
Lung transplant	39 (36%)
Both	14 (13%)
What is your role in your practice?	
Advanced Practice Provider	12 (11%)
Nurse	18 (17%)
Physician	70 (65%)
Other	8 (7%)
Geographical location	
Northeast	22 (23%)
Midwest	28 (29%)
South	17 (18%)
West	10 (10%)
I do not practice in the United States	19 (20%)
(Missing)	12

^1^ n (%).

**Table 2 ijerph-20-04805-t002:** Prevalence of Violence in Heart and Lung Transplantation Programs.

Survey Item	N = 108 ^1^
Has any member of your team (including yourself) experienced a threat of physical violence from a patient or family?	
No	63 (58%)
Yes	45 (42%)
Has any member of your team (including yourself) experienced physical violence from a patient or family?	
No	90 (88%)
Yes	12 (12%)
(Missing)	6
How do you feel physical violence or the threat of physical violence in your practice is changing over time?	
Decreasing	3 (3.2%)
Same	63 (67%)
Increasing	28 (30%)
(Missing)	14
Do you feel safe at work?	
Never	2 (2.1%)
Some of the time	1 (1.1%)
Most of the time	45 (48%)
Always	46 (49%)
(Missing)	14
Do you know another provider who has experienced a threat of physical violence from a patient or family?	
No	47 (45%)
Yes	58 (55%)
(Missing)	3
Do you know another provider who has experienced physical violence from a patient or family?	
No	77 (75%)
Yes	26 (25%)
(Missing)	5

^1^ n (%).

**Table 3 ijerph-20-04805-t003:** Prevalence of Violence by Provider Location (Domestic vs. International).

Survey Item	US, N = 77 ^1^	International, N = 19 ^1^	*p*-Value ^2^	Cramer’s V
Has any member of your team (including yourself) experienced a threat of physical violence from a patient or family?			0.026	0.204
No	39 (51%)	15 (79%)		
Yes	38 (49%)	4 (21%)		
Do you know another provider who has experienced a threat of physical violence from a patient or family?			0.030	0.201
No	30 (41%)	13 (68%)		
Yes	44 (59%)	6 (32%)		

^1^ n (%). ^2^ Pearson’s Chi-squared test.

**Table 4 ijerph-20-04805-t004:** Prevalence of Violence among U.S. Providers by Region.

Survey Item	Northeast, N = 22 ^1^	Midwest, N = 28 ^1^	South, N = 17 ^1^	West, N = 10 ^1^	*p*-Value ^2^	Cramer’s V
Has any member of your team (including yourself) experienced physical violence from a patient or family?					0.010	0.334
No	20 (91%)	25 (93%)	16 (100%)	6 (60%)		
Yes	2 (9.1%)	2 (7.4%)	0 (0%)	4 (40%)		

^1^ n (%). ^2^ Pearson’s Chi-squared test.

**Table 5 ijerph-20-04805-t005:** Prevalence of Violence by Provider Role.

Survey Item	Advanced Practice Provider N = 12 ^1^	Nurse N = 18 ^1^	Physician N = 70 ^1^	Other N = 8 ^1^	*p*-Value ^2^	Cramer’s V
Has any member of your team (including yourself) experienced a threat of physical violence from a patient or family?					<0.001	0.366
No	3 (25%)	6 (33%)	46 (66%)	8 (100%)		
Yes	9 (75%)	12 (67%)	24 (34%)	0 (0%)		
Has any member of your team (including yourself) experienced physical violence from a patient or family?					0.014	0.275
No	8 (67%)	13 (76%)	61 (94%)	8 (100%)		
Yes	4 (33%)	4 (24%)	4 (6.2%)	0 (0%)		
Do you feel safe at work?					0.199	0.108
Never	0 (0%)	0 (0%)	2 (3.3%)	0 (0%)		
Some of the time	0 (0%)	0 (0%)	1 (1.7%)	0 (0%)		
Most of the time	5 (45%)	13 (87%)	23 (38%)	4 (50%)		
Always	6 (55%)	2 (13%)	34 (57%)	4 (50%)		
Do you know another provider who has experienced a threat of physical violence from a patient or family?					0.005	0.305
No	2 (18%)	3 (18%)	36 (52%)	6 (75%)		
Yes	9 (82%)	14 (82%)	33 (48%)	2 (25%)		

^1^ n (%). ^2^ Pearson’s Chi-squared test.

## Data Availability

The data presented in this study are available on request from the corresponding author. The data are not publicly available due to participant privacy concerns.

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
