# Peer review of "Prevalence of Violence against Providers in Heart and Lung Transplant Programs"

_ijerph, 2023, doi:10.3390/ijerph20064805_

Round 1

Reviewer 1 Report

The topic of violence (verbal or physical) towards healthcare workers is very interesting, especially because we are living interesting times.

I find the paper appropriate for publishing with a couple of suggestions maybe for future studies. First of all future studies must take into account the gender of the healthcare worker, since women can be seen by aggressive patients or family members as more susceptible to intimidation, and the risk for violence should be quantified accordingly. Also, another bias to be taken into account is indeed the tracking of the violent events, to make sure that the same event is not being reported twice. 

Overall, this article is very interesting for the readers.

Author Response

Reviewer 1

The topic of violence (verbal or physical) towards healthcare workers is very interesting, especially because we are living interesting times.

CONCLUSION

I find the paper appropriate for publishing with a couple of suggestions maybe for future studies.

1)     First of all future studies must take into account the gender of the healthcare worker, since women can be seen by aggressive patients or family members as more susceptible to intimidation, and the risk for violence should be quantified accordingly.

2)     Also, another bias to be taken into account is indeed the tracking of the violent events, to make sure that the same event is not being reported twice.

Overall, this article is very interesting for the readers.

Yes, violence against health care workers and in general is one facet of many upheavals that we as a global society are experiencing.

1)     We agree that violence may present differently depending on provider gender. As noted in our Limitations section, because we did not ask for demographic information from participants, we were unable to perform subgroup analyses by gender. Of note, research studies do not generally report incidents of violence separately based on gender (Sahebi et al., 2022). However, to emphasize the importance of accounting for provider gender, we added this sentence to the introduction section: “There is also a need to collect different data elements (e.g., provider gender) and harmonize validated data collection tools/instruments to comprehensively study workplace violence.”

2)     We agree, and have described the need for better tracking of violent events in our paper, including adding this point to the Conclusion section.

Thank you.

Reviewer 2 Report

abstract: line 27, is the 42% representative of forty-five participants?

Results: line 141; please indicate that of 42% who have experienced physical violence about 49% are from the U.S and then further show what the study is saying; because if you leave it hanging like that it is misleading, one might not know the real percentage.

It would also be nice to see if the violets are gender based or not. Nurses and advanced practice providers are the one at risk, and there is a need for psychological support as such behaviour can affect their work ethics or dedication to serve patients. 

conclusion: revise conclusion by sumarising results of affected staff. it would be good if affected staff can be supported and have staff working together to reduce the risk, and patients should be involved to also take responsibilities through counselling. 

If consent form, data collection tools and ethical approvals are not attached please attach them.

Author Response

Reviewer 2

Reviewer comments and suggestions:

ABSTRACT

1)     line 27, is the 42% representative of forty-five participants?

1)     Yes. We analyzed a sample of 108 respondents and 45 of those, or 42%, reported threats of physical violence. This summary statistic is reported in Table 2 and presented in section 3.2.

RESULTS

1)     line 141; please indicate that of 42% who have experienced physical violence about 49% are from the U.S and then further show what the study is saying; because if you leave it hanging like that it is misleading, one might not know the real percentage.

2)     It would also be nice to see if the violets are gender based or not. Nurses and advanced practice providers are the one at risk, and there is a need for psychological support as such behaviour can affect their work ethics or dedication to serve patients.

1)     We apologize for any confusion, but that is not the correct interpretation of the results, as percentages in our tables are computed column-wise. To address this potential confusion, the “Overall” column has now been removed from table 3. Tables 3, 4 and 5 are now stratified simply by their variable of interest.

2)     The survey was intentionally designed to be brief in order to facilitate uptake, and excluded most personally identifiable information about the respondents. The Limitations section acknowledges that because of missing demographic characteristics, the role of gender in workplace violence could not be assessed. We recognize the reviewer’s concern, and we propose to return to the issue in future work. For now, we note that available data suggests that gender might not play a significant role in experiences related to violence across the entire career of health care providers. Interestingly, longitudinal research conducted in Norway shows that male providers are more likely to experience threats of violence rather than their female colleagues, but that difference tends to disappear for providers with more years of experience (see Nøland ST, Taipale H, Mahmood JI, Tyssen R. Analysis of Career Stage, Gender, and Personality and Workplace Violence in a 20-Year Nationwide Cohort of Physicians in Norway. JAMA Network Open. 2021;4(6):e2114749. doi:10.1001/jamanetworkopen.2021.14749). Our limitations section has been edited to include this point and the relevant reference.

CONCLUSION

1.     revise conclusion by sumarising results of affected staff. it would be good if affected staff can be supported and have staff working together to reduce the risk, and patients should be involved to also take responsibilities through counselling.

1)     We appreciate this suggestion and have summarized our results in the Conclusion section and have added that future research should include preventive strategies to reduce the risk of workplace violence.

ATTACHMENTS

1)     If consent form, data collection tools and ethical approvals are not attached please attach them.

2)     We have attached the following documents:

Consent form

IRB approval

Survey

Recruitment script

Recruitment QR code

Recruitment study information sheet

Reviewer 3 Report

I congratulate the authors for the topic chosen in their manuscript regarding violence against health workers at work, an extremely sensitive topic with many medical, legal, socio-human, economic and other implications.

I also note the originality of this article, which focuses on workplace violence against the members of the heart-lung transplant teams.

Clear, scientifically relevant manuscript, with a design that methodically follows rigorous statistical analysis, scientifically argued results and conclusions.

In the future, I recommend the authors to deepen the researched topic in 2 directions: 1) the analysis of questionnaires addressed to patients from the transplant program, with targeted questions addressed to the topic for the early identification of patients with impulsive-aggressive valences (including the trigger factors, the educational environment, but and tools to combat the triggering mechanisms - these are just a few examples, the list being much more complex) and 2) analysis of the sensitive indicators resulting from the risk assessment and the proposal of corrective measures, according to the implemented local protocols.

Author Response

Reviewer 3

Reviewer comments and suggestions:

I congratulate the authors for the topic chosen in their manuscript regarding violence against health workers at work, an extremely sensitive topic with many medical, legal, socio-human, economic and other implications.

I also note the originality of this article, which focuses on workplace violence against the members of the heart-lung transplant teams.

Clear, scientifically relevant manuscript, with a design that methodically follows rigorous statistical analysis, scientifically argued results and conclusions.

Thank you. We agree that this topic is important and has implications for future research on many levels and in our revised submission we have broadened our Conclusion section to include several of them.

Thanks. Your comment and those of the other reviewers led to our revision of the Introduction section in order to highlight the originality of our study in that it fills a gap with regard to workplace violence in this particular hospital setting.

CONCLUSION/FUTURE DIRECTIONS

In the future, I recommend the authors to deepen the researched topic in 2 directions:

1) the analysis of questionnaires addressed to patients from the transplant program, with targeted questions addressed to the topic for the early identification of patients with impulsive-aggressive valences (including the trigger factors, the educational environment, but and tools to combat the triggering mechanisms - these are just a few examples, the list being much more complex) and

2) analysis of the sensitive indicators resulting from the risk assessment and the proposal of corrective measures, according to the implemented local protocols.

Thank you for these recommendations for future research. In response, we have made the following revisions to our manuscript:

1)     We agree that designing interventions to mitigate workplace violence is complex. Moreover, the evidence regarding how successful current interventions are at reducing violence is of poor quality (Spelten et al., 2020). Based on your suggestion, we have included the recommendation that more research is needed to develop patient-facing tools to minimize distress and maximize the outcomes of those awaiting heart and lung transplants. This recommendation can be found in the Behavioral Modification and Safety Plan section of our Discussion section.

2)     We have revised the Conclusion section to include several strategies for violence prevention.

Reviewer 4 Report

Although the paper is relatively well-written, there are a number of issues that should be addressed.

The Introduction concisely summarizes the rationale and purpose.  However, I did not see a Literature Review as to what we know and do not know about violence against transplant programs in general or against health care providers. Some of this information is in the Discussion section.  There is some disconnect between the results and the Recommendations.  The Recommendations are not referenced and it is not apparent that they flow from the study results or other literature.

The authors should provide information as to how the ten questions were developed and validated. I note that the concept of “physical violence” is not defined nor is “threat.”  Were the concepts/explanations provided to the respondents? Question three in Table 2 asks about feeling physical violence or the threat of physical violence – these are two distinctly different concepts and normally require two separate questions.  Also, what is meant by “feeling safe”?

As to the sample of respondents, can the authors provide an estimate of the total number of potential respondents? Is there an implied or actual response rate?

The authors do not need to present the full Tables 3, 4 and 5. With so few significant results, I would present only those results. The authors should double check the obtained Chi-square values and probabilities for accuracy.  For those tables that are significant the phi coefficient. Cramer’s V or some other measure of association would assist in the interpretations.  I am struck by the observation that there is no consistent pattern of significant results across the three tables.

Author Response

Reviewer 4

Reviewer comments and suggestions:

Although the paper is relatively well-written, there are a number of issues that should be addressed.

Thank you for your constructive criticism of our paper. We have addressed your concerns as indicated below.

INTRODUCTION

1.     The Introduction concisely summarizes the rationale and purpose. However, I did not see a Literature Review as to what we know and do not know about violence against transplant programs in general or against health care providers. Some of this information is in the Discussion section.

1.     We have updated the literature review to include more recent references as well as pulled in those from the Discussion section.

We have revised the Introduction section to better reflect the gaps in current knowledge about workplace violence in general, and specifically in heart and lung transplant programs.

DISCUSSION

1) There is some disconnect between the results and the Recommendations. The Recommendations are not referenced and it is not apparent that they flow from the study results or other literature.

1)     We have revised our Discussion section so that the recommendations provided for transplant programs to mitigate workplace violence are based on either our study results, our experience as heart and lung transplant providers, and/or the literature.

METHODS

1)     The authors should provide information as to how the ten questions were developed and validated. I note that the concept of “physical violence” is not defined nor is “threat.” Were the concepts/explanations provided to the respondents?

2)     Question three in Table 2 asks about feeling physical violence or the threat of physical violence – these are two distinctly different concepts and normally require two separate questions. Also, what is meant by “feeling safe”?

3)     As to the sample of respondents, can the authors provide an estimate of the total number of potential respondents? Is there an implied or actual response rate?

1)     We adopted the following definition of workplace violence: “Incidents where staff are abused, threatened or assaulted in circumstances related to their work, including commuting to and from work, involving an explicit or implicit challenge to their safety, well-being, or health.” It was sourced from the work of Clarkin and colleagues* and further discussed in the systematic review published by Liu and colleagues**. The introduction of the manuscript has been edited to include the above definition, as well as the relevant references.

2)     The first two questions reported in Table 2 addressed the respondents’ experiences with threats of physical violence and actual physical violence, respectively. Question three aimed to assess respondents’ perceptions regarding a possible increase/decrease of either of those experiences. The survey could have been designed to separately assess respondent perception of change over time of the two items, but we opted for simplicity.

Question four refers to the feeling of safety and was designed to be interpreted by the respondent. The aim was to decouple the frequency of threats/physical violence from the perceptions of the respondents. As we elaborate in section 4.2: “our respondents reported feeling safe at work all or most of the time, despite reporting a high prevalence of violence or threats of violence. More research is needed to understand the reasons for this finding. Specifically, it is possible that healthcare workers have come to accept violence as another facet of their many workplace challenges or that their feelings of safety result from their institutions having adequate protective measures in place.”

3)     In the manuscript we do not attempt to report the survey response rate. This is mainly due to the informal recruiting strategy adopted during the ISHLT annual meeting. In section 2.1 we clarify the methods with the following statements: “Participants were drawn from a convenience sample of conference attendees: four providers from the research team approached conference attendees gathered in small groups between sessions and promoted the study. Potential respondents were provided with a brief explanation of the goal of the study and given access to the online survey via a quick response (QR) code. Due to the informal recruiting strategy, the research team did not attempt to keep track of the number of attendees they interacted with during the conference. All respondents provided informed consent prior to participation.”

* Wynne R, Clarkin N, Cox T, et al. Guidance on the prevention of violence at work. Luxembourg: European Commissions, DG-V, 1997.

** Liu J, Gan Y, Jiang H, et al. Prevalence of workplace violence against healthcare workers: a systematic review and meta-analysis. Occup Environ Med. 2019;76(12):927-937. doi:10.1136/oemed-2019-105849

RESULTS

1)     The authors do not need to present the full Tables 3, 4 and 5. With so few significant results, I would present only those results.

2)     The authors should double check the obtained Chi-square values and probabilities for accuracy.

3)     For those tables that are significant the phi coefficient. Cramer’s V or some other measure of association would assist in the interpretations.

4)     I am struck by the observation that there is no consistent pattern of significant results across the three tables.

1)     Thank you for this suggestion. Tables 3, 4 and 5 have been edited to present only the summaries and tests relative to the survey items described in the Results section.

2)     The Chi-square values and probabilities have all been checked for accuracy and presentation of all tests in the tables has been standardized to 3 decimal digits.

3)     Since many of the tests involve at least a categorical variable with more that 2 levels, Cramer’s V was selected to measure the strength of the association. A new column has been added to Tables 3, 4 and 5 with this measure.

4)     We appreciate this comment but note that since geographic location and provider role are characteristics not related to each other, the lack of a pattern is not unexpected.

Round 2

Reviewer 4 Report

I appreciate the authors' attempt at revising the manuscript. Clearly, they  made significant revisions.  Explanations with regard to question wording and intent are presented as well as a description of how data were collected. I remain disappointed in the quality of the survey form and the methods to collect data.  I do not understand if the respondents were explained the concepts of referenced in my comments. The answer to the comment concerning the mixing of concepts "for simplicity" is not convincing. that the results were not unexpected gives me concern as to the overall rigor of the conceptualization and conduct of the research.

Author Response

Comment

Response

INTRO TO REVIEWER 4

We thank you the opportunity to respond to your comments.

Please note the following:

Reviewer 4

I appreciate the authors' attempt at revising the manuscript. Clearly, they made significant revisions. Explanations with regard to question wording and intent are presented as well as a description of how data were collected.

Thank you for your previous comments. Your constructive criticisms helped us improve the manuscript.

I remain disappointed in the quality of the survey form and the methods to collect data.  I do not understand if the respondents were explained the concepts of referenced in my comments. The answer to the comment concerning the mixing of concepts "for simplicity" is not convincing. that the results were not unexpected gives me concern as to the overall rigor of the conceptualization and conduct of the research.

Thank you for the opportunity to provide clarification. No, we did not define or elaborate on the concepts of ‘physical violence,’ ‘threat of physical violence,’ or ‘feeling safe’ as part of our survey. We did not mean to imply a lack of rigor with our comment on simplicity, but rather that we had intended to keep the survey brief.

Our recruitment message indicated that we were ‘conducting a study to understand heart transplant and lung transplant providers’ perceptions of and experiences with safety threats in their workplace, specifically related to threats from patients and their family members.’ If the editors think this information would be helpful, we are happy to include the consent form as another appendix, but as we do not think that this addition contributes greatly to this submission, we have not uploaded it with this resubmission.

We do agree that it would be useful to separate and define those concepts in any further research we would undertake on violence in the workplace.